

# Formation and evolution of Tar Balls from Northwestern US wildfires

Arthur J. Sedlacek III[1], Peter R. Buseck[2], Kouji Adachi[3], Timothy B. Onasch[4], Stephen R. Springston[1], and Lawrence Kleinman[1]

[1]Environmental and Climate Sciences, Brookhaven National Laboratory, U.S.A.
[2]School of Earth and Space Exploration & School of Molecular Sciences; Arizona State University, U.S.A.
[3]Atmospheric Environment and Applied Meteorology Research Department, Meteorological Research Institute, Japan
[4]Aerodyne Research Inc., Billerica MA, U.S.A.

*correspondence to*: Arthur J. Sedlacek (Sedlacek@bnl.gov)

**Abstract.** Biomass burning is a major source of light-absorbing black and brown carbonaceous particles. Brown carbon is a poorly characterized mixture that includes tar balls (TBs), a type of carbonaceous particle apparently unique to biomass burning. Here we describe the first atmospheric observations of the formation and evolution of TBs from forest fires. Aerosol particles were collected on TEM grids during aircraft transects at various downwind distances from the Colockum Tarp wildland fire. TB mass fractions, derived from TEM and *in-situ* measurements, increased from <1% near the fire to 31-45 % downwind, with little change in TB diameter. Single-scattering albedo determined from scattering and absorption measurements increased slightly with downwind distance. Similar TEM and SSA results were observed sampling multiple wildfires. Mie calculations are consistent with weak light absorbance by TBs (m=1.56 – 0.02i) but not consistent with order-of-magnitude stronger absorption observed in different settings. The field-derived TB mass fractions reported here indicate that this particle type should be accounted for in biomass-burn emission inventories.

## 1. Introduction

Wildfires are major sources of organic- and black-carbon aerosol particles (de Gouw and Jimenez 2009; Bond et al., 2013; Andrea and Rosenfeld, 2008; Park et al., 2007) and tar balls (TBs), for reasons that will become apparent, are under-recognized constituents of wildfire emissions. Here we focus on the production of atmospheric TBs from wildfires in the northwestern United States and agricultural burns in the lower Mississippi Valley.

To help quantify the contribution of biomass burning (BB) to aerosol radiative forcing, the Department of Energy's Atmospheric Radiation Measurement (ARM) program carried out an aircraft-based field campaign (BBOP: Biomass Burning Observation Project; BBOP, 2015) that targeted the near-field evolution (< 5 hrs) of BB aerosol particles. This campaign focused on wildfires in the northwestern United States in the summer and agricultural burns in the lower Mississippi Valley in the fall (Figure 1). The present paper describes the wildfire production of tar balls (TBs).



Tar balls (Pósfai et al., 2003; Pósfai et al., 2004; Li et al., 2003) are a type of carbonaceous particle that is characterized by near sphericity, high-viscosity, absence of graphitic fine structure, and resistance to electron-beam radiation, that appear to be the near exclusive byproduct of some types of biofuel combustion/pyrolysis and wildfires (Tóth et al., 2014; Tivanski et al., 2005; Hand et al., 2005; Cofer et al., 1988). Detection of TBs is currently limited to electron microscopy (transmission electron microscopy [TEM; Pósfai et al., 2004; Adachi et al., 2011], secondary electron microscopy [SEM; Chakrabarty et

al., 2006; Chakrabarty et al., 2010; China et al., 2013] and scanning transmission X-ray spectroscopy [STXM; Tivanski et al., 2007]).

    The origin of TBs is controversial. Suggestions include formation through secondary gas-to-particle polymerization during pyrolysis (Pósfai et al., 2004) and the rapid heating of directly injected 'tar' droplets into the atmosphere (Tóth et al. (2014). We find that they form during volatile loss and maturation in the emission plume.

In some wildfire samples TBs are the dominant particle type collected on microscopy sample grids, out-numbering black carbon particles by a factor of ten (Hand et al., 2005; China et al., 2013). Clearly, TB formation and their microphysical/optical properties must be understood so that their radiative forcing contribution can be properly assessed and incorporated into climate models.

    Using aircraft-based, pseudo-Lagrangian sampling strategy in which samples of different plume ages were collected onto

separate TEM grids, we report on the direct observation of TB formation downwind of the northwestern U.S. Colockum Tarps wildfire[1] near Ellensburg, WA. Flight tracks were perpendicular to the wind-advected fire plume with one upwind and six downwind transects repeated 2 or 3 times each. Our analysis combines the ratio of TB/ns-soot[2] derived from TEM analysis with non-refractory particulate matter (NR-PM) from the Soot Particle Aerosol Mass Spectrometer (SP-AMS; Aerodyne Research, Inc.) and refractory black carbon (rBC) loadings from the Single Particle Soot Photometer (SP2;

Droplet Measurement Technologies). We observe a steady increase in the TB number and volume fraction with plume age. From the combined set of instruments, we obtain the first estimate of the particulate mass fraction of TBs in hours-old plumes: (31-45 %). A comparison of single-scattering albedo (SSA) from optical measurements compared with Mie calculations for limiting values of the TB refractive index (Alexander et al., 2008; Hoffer et al., 2016) suggest that TBs exert a cooling influence on such smoke plumes.

**2. Methods**

---

[1] http://inciweb.nwcg.gov/incident/3567/

[2] "Black carbon" has several meanings. Here we adopt the usage suggested by Buseck et al. (2014) and refer to ns-soot for situations where electron microscopy has shown the carbon to consist of small fractally arranged spheres, i.e., a common sub-group of black carbon.



## 2.1 Aircraft Platform

The BBOP field campaign comprised two deployments – a wildfire deployment staged in the Pacific Northwest (Richland, WA) from July – mid-September 2013 and an agricultural (controlled) burn deployment staged in the lower Mississippi Valley (Memphis, TN) during October 2013 (Figure 1). Schmid et al. (2014) described the research platform used during
the BBOP field campaign. The aircraft was equipped with an isokinetic aerosol inlet on the starboard side of the aircraft and a dedicated trace gas inlet on the port side. Ambient aerosol samples are brought to individual instruments via conductive tubing while trace gas sampling utilized PFA tubing and/or T316 stainless steel. All instruments were time synced prior to every flight. The Gulfstream-1 (G-1) aircraft was outfitted for four measurement classes (i) microphysical properties, (ii) optical properties, (iii) trace gases, and (iv) radiation. Below are measurement details for those instruments in the present
study.

## 2.2 Instruments

### 2.2.1 Electron Microscopy.

Aerosol particles were collected onto TEM lacey-carbon substrate grids (Type 01881, Ted Pella, lacey Formvar/carbon, 200 mesh Cu grids) using a two-stage aerosol impactor sampler (AS-16W, Arios) that could collect 16 TEM grid samples per
flight. Sampling times ranged from 1 to 30 minutes depending on particle loading. The lower and upper 50% cutoff aerodynamic diameters for the samples used in this study are approximately 100 nm and 700 nm, respectively.

Particle analysis was carried out using a 120kV transmission electron microscope (JEM-1400, JEOL) equipped with an energy-dispersive X-ray spectrometer (EDS; Oxford instruments). A total of ~ 3300 particles were examined from representative images of the 16 Colockum Flats TEM grids in order to determine TB number fractions. A Gatan 628 single-
tilt holder was used for TB heating experiments (Adachi et al., 2017). Ns-soot volumes as determined from 2D TEM images using fractal parameters from Adachi et al (2010) had the same qualitative trends derived from SP2 measurements of refractory black carbon (rBC).

### 2.2.2 Trace Gas and Optical Measurements

The oxides of nitrogen (e.g., NO, $NO_x$, $NO_y$) were measured with a three-channel chemiluminescent instrument (Air Quality
Design, Model 1). $NO_2$ was converted to NO by a high intensity LED, and $NO_y$ was reduced to NO by a 350º C Mo converter mounted at the tip of an inlet outside of the aircraft. CO was measured by intra-cavity off-axis spectroscopy (Los Gatos, Model 907).

Aerosol scattering at 700, 550, and 450 nm was determined using a TSI 3563 integrating nephelometer. Correction factors for truncation and non-cosine response were applied based on the method of Anderson and Ogren (1998) for sub-micrometer





particles. A single-wavelength photothermal interferometer (PTI) was used to acquire *in situ* aerosol absorption at 532 nm (Sedlacek and Lee, 2007; Cross et al., 2010).

### 2.2.3 Aerosol Mass

### 2.2.3.1 refractory black carbon

    Mass concentrations of refractory black carbon (rBC) particles with volume equivalent diameters between ~80 and 500 nm,

assuming a rBC density of 1.8 g cm$^{-3}$, were measured via laser-induced incandescence (Single Particle Soot Photometer (SP2) by Droplet Measurement Technologies) and calibrated with fullerene soot (Alfa Aesar; stock: #40971. lot#: L18U002).

### 2.2.3.2 Aerosol Mass Spectrometry

    An Aerodyne SP-AMS (Onasch et al., 2012) was used to measure the non-refractory particulate matter (NR-PM), including

organic and inorganic aerosol. Standard AMS measurement uncertainty is estimated to be 25% as discussed by Canagaratna et al. (Canagaratna et al., 2007). The SP-AMS uses two methods for vaporizing particles. As in a conventional AMS, a resistively heated tungsten vaporizer flash volatilizes non-refractory aerosol in a vacuum followed by electron ionization and detection via mass spectrometry (Canagaratna et al., 2007). With dual vaporizers, light absorbing refractory aerosol are vaporized by a Nd:YAG laser, after which the particle beam encounters the tungsten vaporizer. The tungsten vaporizer was

calibrated with ammonium nitrate and the laser vaporizer was calibrated with atomized Regal black multiple times during the BBOP study following established protocols (Onasch et al., 2012; Canagaratna et al., 2007; Willis et al., 2014). Onasch et al. (2012) discuss the complications that may occur with dual vaporizers with respect to collection efficiencies (CEs). Lee et al. (2015) observed the CE of ambient NR-PM increased when both vaporization sources were used. During the BBOP campaign, the SP-AMS was operated using the tungsten vaporizer with or without laser vaporization; laser-on or laser-off

modes, respectively, and also observed higher NR-PM for laser-on conditions. Changes between these two modes were done from flight to flight, from one plume transect to the next, and sometimes on shorter time scales. The CE for the SP-AMS operating in laser-off mode was determined to be 0.5 from 2 intercomparison flights with a surface-based AMS at Mount Bachelor Observatory (Collier et al., 2016) and correlations with scattered light measured at 550 nm, where the mass-specific scattering coefficients (MSC) are expected to be near 3.6 ± 1.2 m$^2$g$^{-1}$ (Hand and Malm 1990). All the data for the

Colockum Tarp fire presented here were collected with the SP-AMS in laser-on mode. In order to determine the appropriate CE for non-refractory aerosol under laser-on conditions, we compared laser-on with laser-off measurements for five similar wildfires measured during BBOP, in which 16 plume transect pairs were sequentially sampled under similar conditions for laser-on and laser-off. Averages of the NR-PM measured by the SP-AMS during these sequential plume transects were normalized by measured CO and/or light scattering and the laser-on to laser-off ratios calculated. Counting each transect



pair as a data point and using the laser-off CE = 0.5 as the presumed ambient NR-PM loadings, we obtain a laser-on CE = 0.76 (1σ = 0.10) with CE values that range from 0.63 to 0.88 for a given transect pair.

## 2.3 Plume Sampling Strategy

For the wildfire flights, a pseudo-Lagrangian sampling protocol was employed in which flight transects orthogonal to the plume direction were conducted at selected distances downwind of the source. The plume age was calculated using
prevailing wind speed and direction together with the assumption that the emission source was constant for all sampling transects.

### 2.3.1 Photochemical Age and Plume Dilution

Photochemical age of the smoke plume was calculated using the ratio of $NO_x$ to $NO_y$ as described by Kleinman et al. (2008). Effects of plume dilution were corrected by normalizing data stream (e.g., scattering, absorption, mass loading) by CO
enhancement - a conserved tracer.

### 2.4 Colockum Tarps Fire

The Colockum Tarps fire sampled on the afternoon of 30-July 2013 is representative of at least eight wildfire plumes studied in BBOP with multiple transects covering ~2 to 4 hours of transport. The fire was first noted on 27 July and was declared 98% contained on August 15[3]. A total of ~80,000 acres, consisting of short grass, timber grass understory, and hardwood
litter were burned. An analysis of the time dependence of gaseous and aerosol constituents shows that the fire is typical of several other wildland fires sampled in the BBOP campaign, the description of which will be the subject of future publications.

Figure 2 shows the aircraft ground track for the afternoon July 30th flight. There are 13 plume crossings and one upwind leg. Each is nearly orthogonal to the plume transport direction, as determined from aircraft wind observations. Locations of
TEM samples are color-coded and labeled in Figure 3 in sequential order (1 to 16) and according to their downwind position (T0 to T6). There are two sets of transects, referred to here as Set A and Set B, each of which cover a sampling distance from the main fire region to ~35 km downwind in 6 steps from A-T1 to A-T6 and B-T1 to B-T6 (Figure 3). TEM sample 1 at location T0 was collected upwind of the Colockum Fire. Plume travel, or aging, times between consecutive transects is approximately 30 minutes, and the aging time probed is ~ 3 hours. Concentration of CO, a common combustion tracer,

---

[3] http://inciweb.nwcg.gov/incident/3567/





during a time series (Figure 3) ranges between approximately 0.2 ppm upwind to 5 ppm over fire hot spots near transects T1 and T2. Peak concentrations decrease with downwind distance due to dilution.

$NO_x$ emitted by forest fires is oxidized to peroxyacetyl nitrate (PAN), organic nitrates, and $HNO_3$, which collectively are detected as $NO_y$ (Fisher et al., 2010; Fisher et al., 2016). A qualitative measure of atmospheric oxidative processing is thereby provided by $-\log_{10}(NO_x/NO_y)$, as described by Kleinman et al., (2008), and its increase with downwind distance

(Figure 4). Little change occurs during the one hour separating the first set of transects (Set A) from the second set (Set B), indicating fire conditions were approximately steady during that time period.

### 2.5 BBOP ARM Data Archive

All data used in the present analysis are publicly available on the Department of Energy ARM data archive (http://www.archive.arm.gov/armlogin/login.jsp). DOE ARM verifies data quality through quality assurance and data quality

checks.

## 3. Results

### 3.1 Tar Ball Production:

Four TEM images of aerosol particles with increasing photochemical age, collected near the active fire (0 hrs) and at three downwind distances, are shown in Figure 5. Four types of carbonaceous particles are identified: (1) solid ns-soot, (2) low-

viscosity organic aerosol (OA), (3) high-viscosity OA, and (4) solid TBs. TBs and ns-soot are stable under conditions used to obtain TEM images and both can be unambiguously identified based on morphology. High-viscosity organic particles are distinguished from solid TBs by particle deformation (Adachi et al., 2017) upon impact with the lacy-carbon grid. Due to the spreading flow of low-viscosity particles and material evaporation during storage or electron-beam analysis, quantification of total OA by TEM is difficult (Adachi et al., 2017).

Over the fire, the TB number concentration is near zero and OA particles dominate. As the plume ages, the TEM images reveal an increase in the number fraction of TBs, with the highest number fraction, 64%, observed in the most aged portion of the plume (A-T6 & B-T6 in Figure 3). In air that has the chemical signature of aged, diluted BB smoke (A-T0 in Figure 3, which was collected upwind of the fire), the number fraction of TBs is 36%. These number fractions are similar to those reported by others (Pósfai et al., 2003; Pósfai et al., 2004; Hand et al., 2005; China et al., 2013).

Since TEM number fractions reported here and elsewhere (Hand et al., 2005; China et al., 2013) are sensitive to the loss of volatile particles, a more useful statistic is the number ratio of TBs to ns-soot, which is 3.8 and 7.6 for the downwind transects 15 and 16 (Figure 3). The TB/ns-soot number ratio in aged, diluted BB air sampled upwind of the fire (i.e., background) is 10.8. These ratios are similar to the number ratio of 10 reported by China et al. (2013).



If the Tóth (2014) mechanism - direct injection of precursor 'tar' droplets followed by rapid heating – were indeed
responsible for TB formation, then the BBOP TEM images should reveal TBs at the source, since that is where the high-temperature zone, necessary for particle solidification, is located. Instead, there is a steady growth in TB/ns-soot mass ratio as the plume ages. This increase indicates that TBs are not directly emitted, but rather form (or transform) in the atmosphere within a few hours from emission. Measurements of TB abundance in smaller fires suggest that rapid changes can occur within 15-minute of atmospheric aging (Adachi and Buseck 2011). Adachi and Buseck (2011) observed 10-fold more TBs in the plumes of more-distant fires compared with nearby fires. In contrast to the present study, fresh and aged smoke samples reported by Adachi and Buseck (2011) were in general from different fires. TEM samples with more TBs showed OA morphological changes consistent with an increase in viscosity. Apparently, atmospheric processing causes some OA particles that start out deformable to solidify (Adachi et al., 2010) and be recognized as TBs in the TEM images. The atmospheric processing leading to TBs may involve dehydration and oligomerization of low-volatility OA (Pósfai et al., 2004; Adachi and Buseck, 2011).

Sampling within a single fire as presented here provides further evidence that TBs are processed primary particles (see Discussion Section on nomenclature). The downwind increase in TB mass is due to an increase in number concentration rather than a change in particle size (green 25[th], 50[th], and 75[th] percentile lines in the lower panel of Figure 6). To accomplish a number increase by condensation would require that particles smaller than the 100-nm sampling cutoff for accurate TEM imaging grow into a detectable size range, a feature not observed in the TEM measured size distributions from which average TB diameter is calculated (upper trace in Figure 6). TEM samples often included background air with varying properties along with the plume being investigated and is likely responsible for the spread observed between the 25[th] and 75[th] percentile lines derived from individual data points (dashed and dot-dashed green lines in Figure 5).

### 3.2 Tar ball Mass Fractions

To date, the only way to definitively identify TBs has been through microscopy. However, issues associated with particle volatility have severely limited evaluations of the radiative forcing contribution by this aerosol type. In response to this limitation, we combine TEM image analysis with SP2 and SP-AMS measurements to derive the first estimates of the mass loadings and mass fractions of TBs in wildland fire plumes. TB mass loadings and the TB mass fractions are derived by combining i) the number ratio of TBs to ns-soot determined from TEM images with ii) speciated aerosol concentration measurements from the SP2 and SP-AMS, averaged over the TEM collection time periods.

In our calculation of the TB mass fractions, we make three significant assumptions: First, TEM grids collect TBs and ns-soot with the same efficiency; Second, ns-soot, as determined by TEM, is the same as rBC measured with an SP2. A similar assumption was recently invoked by Adachi et al. (Adachi et al., 2016). For convenience, both quantities will be called "*soot*", and; Third, the SP-AMS collection efficiency (CE) for TBs is the same for other NR-PM. Adachi et al (2017)



studied the thermal vaporization of TBs by heating BBOP samples to 600°C and found them to be less volatile than most

OA, suggesting that TBs may exhibit a different CE in the SP-AMS than other OA (see supplemental). However, given the

current lack of chemical information on TBs in general, TB chemical signatures cannot be deconvolved from SP-AMS OA

mass spectra. Therefore, calculations are done assuming TBs have the same CE as other OA.

Ambient particulate mass concentrations (e.g. μg m⁻³ at STP) of TBs, soot, non-refractory inorganics, and non-refractory

organics are denoted as $M_{TB}$, $M_{soot}$, $M_{IN}$, and $M_{ORG}$, respectively. TB fractional contribution to total mass is denoted by $f_{TB}$

and we define R as the ratio $M_{TB}/M_{soot}$ derived from TEM. We have:

$$M_{total} = M_{TB} + M_{soot} + M_{IN} + M_{ORG} \qquad \text{(1a)}$$

$$f_{TB} = {M_{TB}}/{M_{total}} \qquad \text{(1b)}$$

$$M_{TB} = RM_{soot} \qquad \text{(1c)}$$

The SP-AMS was used to measure $M_{AMS}$, the non-refractory component of $M_{total}$.

$$M_{AMS} = M_{IN} + M_{ORG} + M_{TB} \qquad . \qquad \text{(2)}$$

Except for a sensitivity calculation[4], the SP-AMS collection efficiency, CE, is assumed to be the same for all non-refractory

species; that is $CE_{ORG} = CE_{IN} = CE_{TB}$. Equation 2 is written with separate contributions to total mass from TBs and organics.

However, in practice, neither $M_{ORG}$ nor $M_{TB}$ can be obtained from Eq. 2 because additional chemical information is needed to

allow the SP-AMS to discriminate tar ball organic signals from other organic particulate matter. Combining Eqs. 1 and 2,

we obtain

$$M_{total} = M_{soot} + M_{AMS} \qquad \text{(3a)}$$

$$f_{TB} = \frac{M_{TB}}{M_{soot} + M_{AMS}} \qquad \text{(3b)}$$

For an error analysis, it is convenient to restate Eq. 3b in terms of R,

$$f_{TB} = \frac{R}{1 + \left(\frac{M_{AMS}}{M_{soot}}\right)} \qquad \text{(4)}$$

---

[4] Sensitivity calculations were conducted with the SP-AMS TB detection efficiency reduced by 30% according to the heating

experiments shown in Figure S2 and by Adachi et al. (2017). The calculated $f_{TB}$ was reduced by 0 to 13% depending on

transect.





To a first approximation, sub-micrometer aerosol mass loadings in wildfire plumes consist of less than 10% ns-soot and inorganic constituents, with the remainder split between OA and TBs. TB/ns-soot number ratios derived from TEM (Adachi et al., 2010) are used to derive volume ratios by combining the observed TB diameters from TEM with the rBC volume equivalent diameters (VEDs) derived from SP2 measurements. As discussed above, this approach assumes that ns-soot, as

determined by TEM, is the same as rBC measured by the SP2. Use of the SP2-derived ns-soot VED is considered superior to a TEM-derived VED since the latter is very sensitive to fractal parameters used (Adachi et al., 2010). These volume ratios can be converted into mass ratios using assumed densities. For a TB density (Alexander et al., 2008) of 1.5 g cm$^{-3}$ and an ns-soot spherule density (Park et al., 2004) of 1.8 g cm$^{-3}$, the TB/soot mass (volume) ratios are 16.3 (19.6) and 39.3 (47.2) for the downwind transects 15 and 16, respectively. Figure 7a shows the averaged TB/soot mass as a function of plume

age.[5]

The calculation of $f_{TB}$ for a typical downwind sample is given in Table 1. A one standard deviation accuracy has been estimated by combining individual measurement errors (listed in Table 1) in quadrature, assuming that they are uncorrelated. It is seen that the accuracy of $f_{TB}$ depends primarily on R, which we estimate has a 50% uncertainty.

Figure 7b shows that TB mass fractions increase with plume age. Over the fire, this mass fraction is close to zero and

increases in the aged plume to 31-45%, or approximately a third of the particle mass in the smoke plume. The error bars in Figure 7b represent the measurement precisions (1σ); the estimated uncertainties in $f_{TB}$ is 61%. These values represent the first field measurements of TB mass fractions. Upwind of the fire, in aged biomass burn smoke, the TB mass fractions are nearly 50% of the ambient PM. Figure 7c shows the ΔOA/ΔCO ratio as a function of plume age. While this ratio remains nearly constant over the first couple of hours of aging, increases in the TB/soot mass ratio and TB mass fractions are

observed. This comparison supports a downwind formation mechanism. Similar observations of minimal change in the ΔOA/ΔCO have been previously reported (Jolly et al, 2012; Jolleys et al., 2014; Zhou et al., 2017).

These estimates for the TB mass fractions in BB plumes are similar to the mass fractions observed for a low-volatility PMF factor (BBOA-3) recently reported by Zhou et al (2017) in smoke from wildfires sampled at Mount Bachelor Observatory (MBO) during the BBOP campaign. TEM samples from overflights of MBO had similarly high TB to ns-soot number ratios

as those reported here.

---

[5] TEM sample 8, collected on transect 6, had only 40 analyzed particles, 39TBs and 1 ns-soot. The high ratio of TBs to ns-soot is consistent with the trend deduced from other samples showing increasing TB concentrations downwind. However, because of the very high uncertainty associated with a single particle, sample 8 was not included in our calculations, leaving TEM samples 15 and 16 to represent transect 6.





## 4. Discussion

### 4.1 Nomenclature

The results presented here raise the question as to how best to describe TBs. Are they processed primary particles, as we have elected to describe them here, or are they secondary particles? The argument for the latter can be found in the atmospheric chemistry community where $SO_2$ is a primary pollutant but its oxidation product, sulfate, is called secondary. Applying similar logic, if emitted organic particles are converted to tar balls via oxidation reactions TBs would be classified as secondary. However, within the aerosol community, secondary organic aerosol (SOA) has exclusively been used to describe particulate matter condensed from the gas phase. We did not observe TBs on the TEM grids exposed over the fire but they were found in increasing numbers downwind. Because we did not observe particle growth (Figure 7c) a gas-to-particle condensation mechanism for TB formation seems implausible. We have therefore elected to label TBs as *processed primary* particles.

### 4.2 Inventory Implications:

Top-down, bottom-up comparisons of satellite-retrieved optical properties (e.g., aerosol optical depth or AOD) with inventory-based optical properties reveal a discrepancy that requires emissions to be scaled up by a factor ranging from 1.5 (Reddington et al., 2016) to over 10 (Bond et al., 2013; Lu et al., 2007; van der Werf et al., 2010; Kaiser et al., 2012; Kopacz et al., 2010; Reid et al., 2009). Our results and those of Adachi et al. (2017) underscore the importance of studying TBs, including their physio-chemical and optical properties, such that new techniques can be developed to identify (e.g. via specific chemical signatures) and accurately account for TBs in bottom-up inventories. Several scenarios are possible, one of which is that TBs may have escaped accounting in wildfire inventories (Urbanski 2014; Reid et al., 2005), which would reduce the top-down/bottom-up discrepancy. Liu and co-workers (2017) recently highlighted the importance of quantifying wildfire and prescribed burn emission factors.

### 4.3 Optical Properties

Inconsistencies in the values of the TB refractive index (Table 2) preclude a useful assessment of the radiative impact of these particles. Aerosol SSA was calculated on the basis of the mass fractions, mass absorption coefficient (MAC) and mass scattering efficiency (MSE) of each species. A mass-weighted average was used as if the aerosol was an external mixture. More complicated schemes were not warranted in the absence of mixing state information. Using the mass fractions determined from Eqn. 3b (4) and the definition of SSA, we obtain

$$SSA = \frac{\sum f_i MSE_i}{\sum f_i MSE_i + \sum f_i MAC_i} \tag{5}$$



where *i* labels the four aerosol components in our calculation.

Comparisons of the SSA derived from optical measurements with Mie calculations that assumed TBs were either weak (1.56 - 0.02i) or strong absorbers (1.67 – 0.27i) are shown in Figure 8 as a function of plume age. When TBs are assumed to be stronger absorbers, as reported by Alexander et al. (2008) and Hoffer et al. (2016), the increase in TB mass fraction as a function of plume age should result in a large decrease in SSA. However, such a decrease is not observed. Instead, there is a slight increase in SSA with age. This comparison indicates that TBs in the Colockum Tarp wildfire are weaker absorbers

than suggested in laboratory studies (Hoffer et al., 2016; Hoffer et al., 2017). Other wildfires sampled in BBOP had comparable TB-to-soot ratios and age trends. SSA's were also similar in magnitude and time trend to the Colockum Tarp wildfire so we believe our observations at least pertain to fires in this region.

MAC and MSE for TBs were determined from Mie calculations based upon literature values of refractive index in Table 2, a log normal fit to the observed size distribution (count median diameter =242 nm, geometric standard deviation = 1.32), and a

TB density of 1.5 g cm⁻³. Results are given in Table 3 along with literature values used for the other aerosol components. Plausible changes to optical parameters other than the MAC of TBs had little effect on the results in Figure 8.

The causes for disparate measurements of TB refractive index have not been identified. We note that high absorptivity TBs made in the laboratory via the Tóth et al. (2014) mechanism, such as those used by Hoffer et al. (Hoffer et al., 2016; Hoffer et al., 2017), are formed differently than the ambient TBs reported here. The other high absorption result in Table 3 reported

by Alexander et al. (2008) is from an EELS (electron energy-loss spectroscopy) spectrum of a collected ambient aerosol, but it is not clear that the TBs analyzed by Alexander have a BB origin. These comparisons suggest there may be variability in TB optical properties dependent upon fuel source, formation mechanism, and, potentially, aging.

**4.4 Radiative Forcing Implications:**

Incorporation of TBs into models of BB (Jacobson, 2014) have been hampered because the only technique for detecting

these particles is single particle microscopy, which is an off-line, intensive technique with sampling issues (e.g., does not quantify volatile aerosol lost in storage or upon electron beam irradiation). Our work shows that TBs can represent a significant fraction of the aerosol mass in some wildfire plumes and that regional effects of TBs over biomass burning dominated regions would be much more significant. Inclusion of TBs also will help to constrain the BC radiative forcing in biomass burning.

**5 Acknowledgements**

The authors gratefully acknowledge Ernie Lewis and Leah Williams for help with laboratory studies of TBs as well as discussions about the AMS collection efficiency in the derivation of the mass fraction expressions. This research was



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





**Table 1: Calculation of the fractional mass of TBs and its uncertainty.**

| Parameter | Value | Fractional uncertainty |
|---|---|---|
| R | 32.5 | 0.5 |
| $M_{AMS}/CE$ | 119 µg m$^{-3}$ | 0.25 |
| $M_{soot}$ | 1.7 µg m$^{-3}$ | 0.25 |
| $f_{TB}$[1] | 0.46 | 0.61 |

[1] Downwind value of $f_{TB}$ in text is an average over TEM samples 8, 15, and 16 and has values of 0.26.





**Table 2: Measured or derived refractive Indices for Tar Balls**

| Refractive Index | TB Source/Analysis Technique | Wavelength (nm) | Reference |
|---|---|---|---|
| 1.67 - 0.27i | ACE-Asia Field campaign <br><br> - Electron Energy Loss Spectroscopy <br><br> (EELS) | 550 | Alexander et al. 2008 |
| 1.84 – 0.21i | Tar-water emulsion <br><br> - Light Absorption | 550 | Hoffer et al., 2016 |
| 1.56 – 0.02i | YACS field campaign <br><br> - OC/EC ratio & Scattering | 632 | Hand et al., 2005 |
| 1.80 – 0.007i | Ponderosa Pine <br><br> - Light Absorption | 532 | Chakrabarty et al., 2006 |
| 1.75 – 0.002i | Alaskan Duff <br><br> - Light Absorption | 532 | Chakrabarty et al., 2006 |





**Table 3 MAC and MSE of aerosol components**

| Component | MAC ($m^2$ $g^{-1}$) | MSE ($m^2$ $g^{-1}$) | Reference |
|---|---|---|---|
| TB (m = 1.67 – 0.27i)[1] | 3.77 | 3.65 | Alexander et al., 2008 |
| TB (m = 1.84 – 0.21i)[1] | 3.63 | 5.35 | Hoffer et al., 2016 |
| TB (m = 1.56 – 0.02i)[1] | 0.47 | 4.49 | Hand et al., 2005 |
| TB (m = 1.80 – 0.007i)[1] | 0.22 | 8.57 | Chakrabarty et al., 2010 |
| Soot | 11.25[2] | 3.2[3] | Bond et al., 2013 |
| PM = Organic + Inorganic | 0.0[4] | 3.5[5] | Briggs et al., 2016 |

1. Refractive index used for Mie calculation

2. Uncoated soot x 1.5 to account for coating

3. Estimated to give an SSA for soot of 0.3 for uncoated soot

4. Minor absorption at 550 nm ignored

5. MSE determined at biomass burns geographically close to our observations. The average value of 3.5 $m^2$ $g^{-1}$, determined for ambient air samples, has been applied to our organic and inorganic fraction.  The error incurred is less than the variability in MSE between fires.





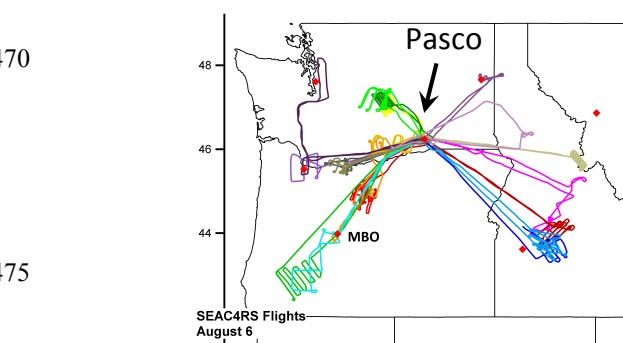
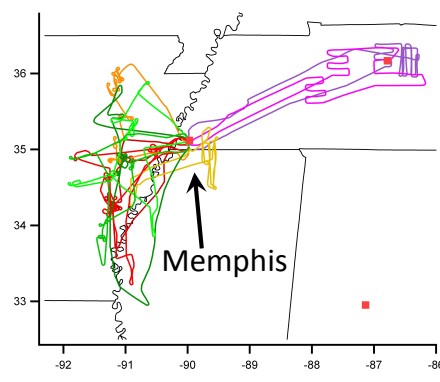

**Figure 1. Compilation of all BBOP flight tracks. The summer deployment was staged out of Pasco (WA) and targeted 17 wildfires in Idaho, Oregon, and Washington. The fall deployment targeted over 24 agricultural burns in the lower Mississippi Valley and was staged out of Memphis (TN). Five research flights targeted urban centers (Seattle, Portland, Spokane, Nashville, Memphis).**





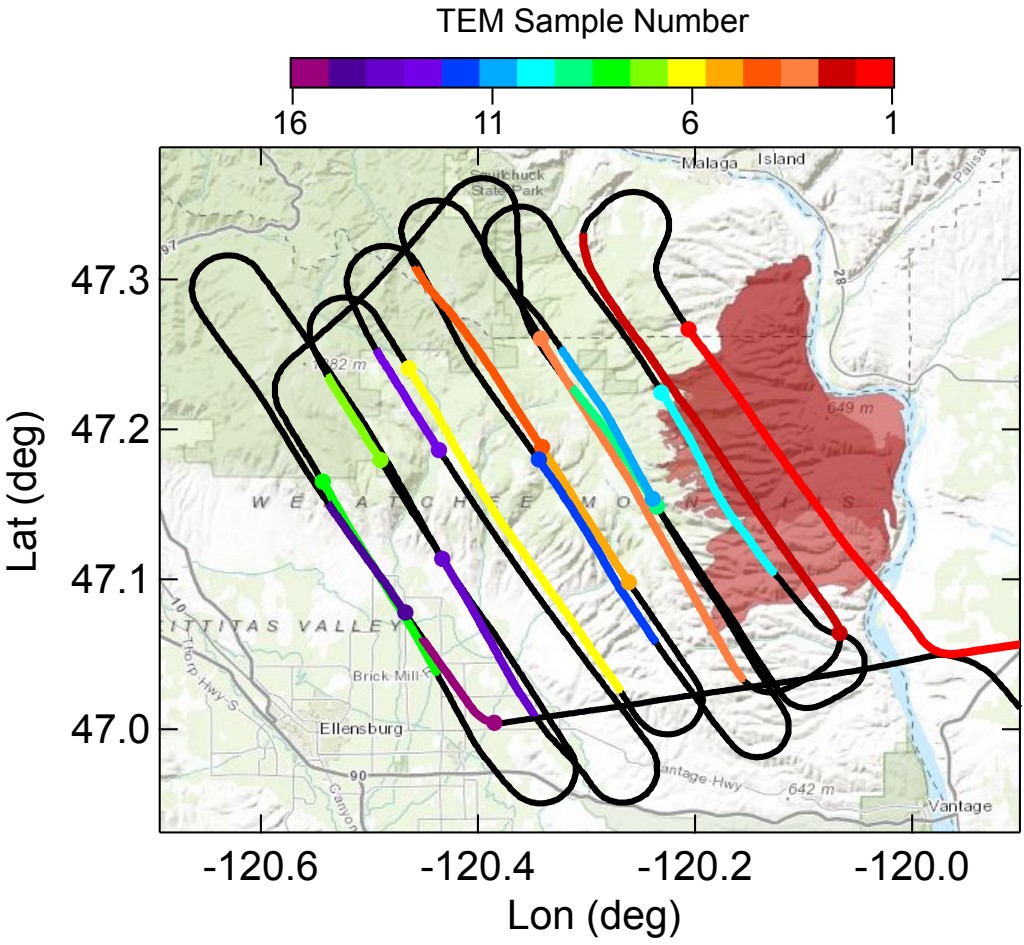

**Figure 2. BBOP G-1 flight track for the Colockum Tarps Fire. Knowing the wind speed and direction, individual transects sample smoke plume at specific ages. Orthogonal transects roughly represent 30-minutes of aging starting from 0 hours over the fire. For more information see Colockum Tarps Fire, 2013. Ground track of aircraft for Flight 730b indicating locations in which 16 TEM grids were sequentially used to collect aerosol samples. Color-coding is used to distinguish TEM sampling. The red trace, from the first transect, represents is upwind of the fire.**






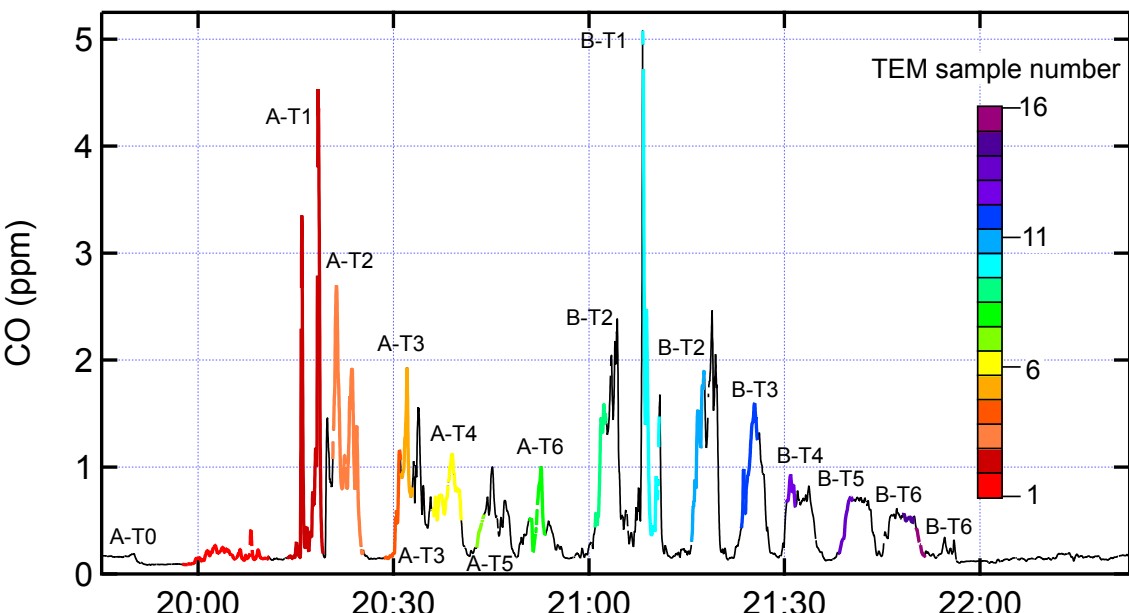

**Figure 3.** **Time series for CO, using same color scheme as Figure 2 to label TEM samples. Two sets of transects, called Set A and Set B, cover a similar sampling distance from just upwind of the main fire region (T0) to ~35 km downwind in 6 steps from T1 to T6. Set A encompasses transects A-T1 to A-T6 and Set B transects B-T1 to B-T6. 16 TEM grids were used in this flight.**





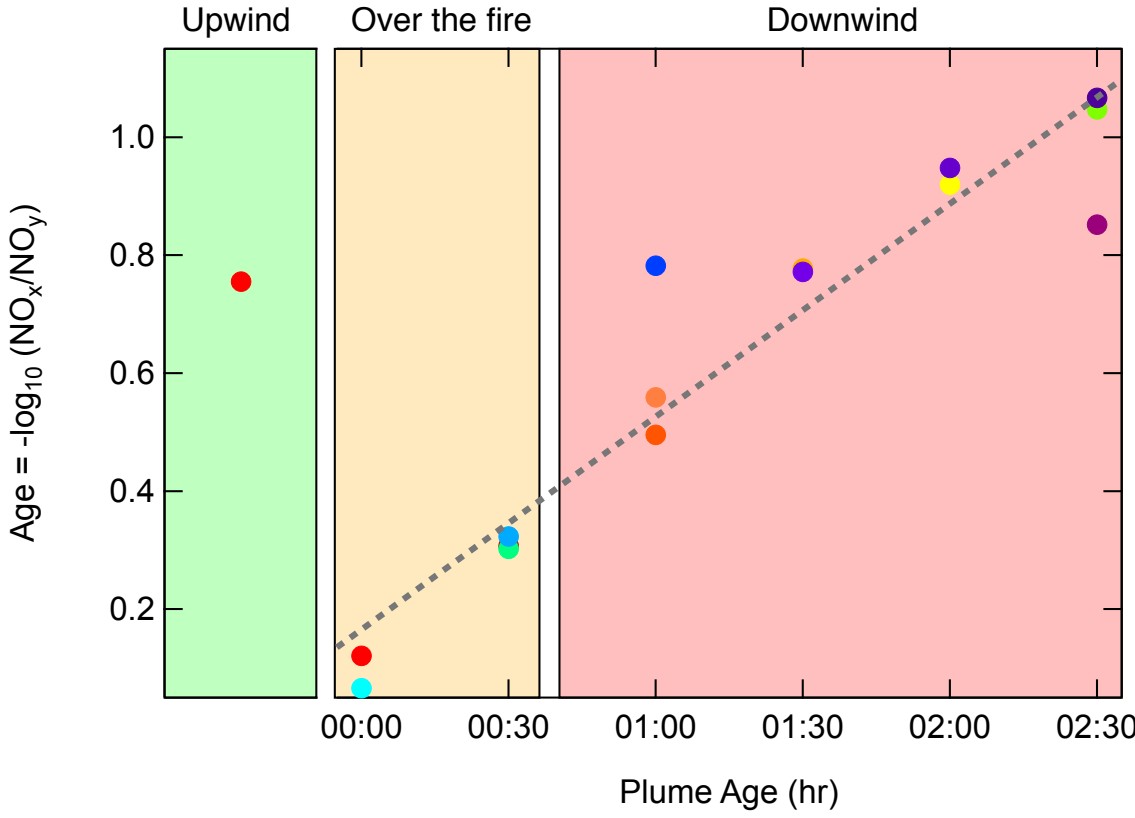

**Figure 4 Photochemical age metric, -Log₁₀(NOₓ/NOᵧ), as a function of plume age. Clean air background subtracted from NOₓ and NOᵧ. Grey dotted line is a linear best fit to the dataset. Colors on the plot correspond to those in Figures 2 and 3.**







**Figure 5: TEM images collected from the Colockum Tarps fire. The spider-web-like strands consist of carbon from the TEM grid. Near the source (0 hrs), low viscosity organic matter predominates and TBs are absent. TBs steadily increase in number fraction as the plume ages from 1 to 2.5 hr, while lower viscosity organic particles decrease in number fraction. Note the change in magnification in the 2.5 hr image. Images for other transects and other wild fires during BBOP exhibit the same change in particle characteristics as a function of plume age.**






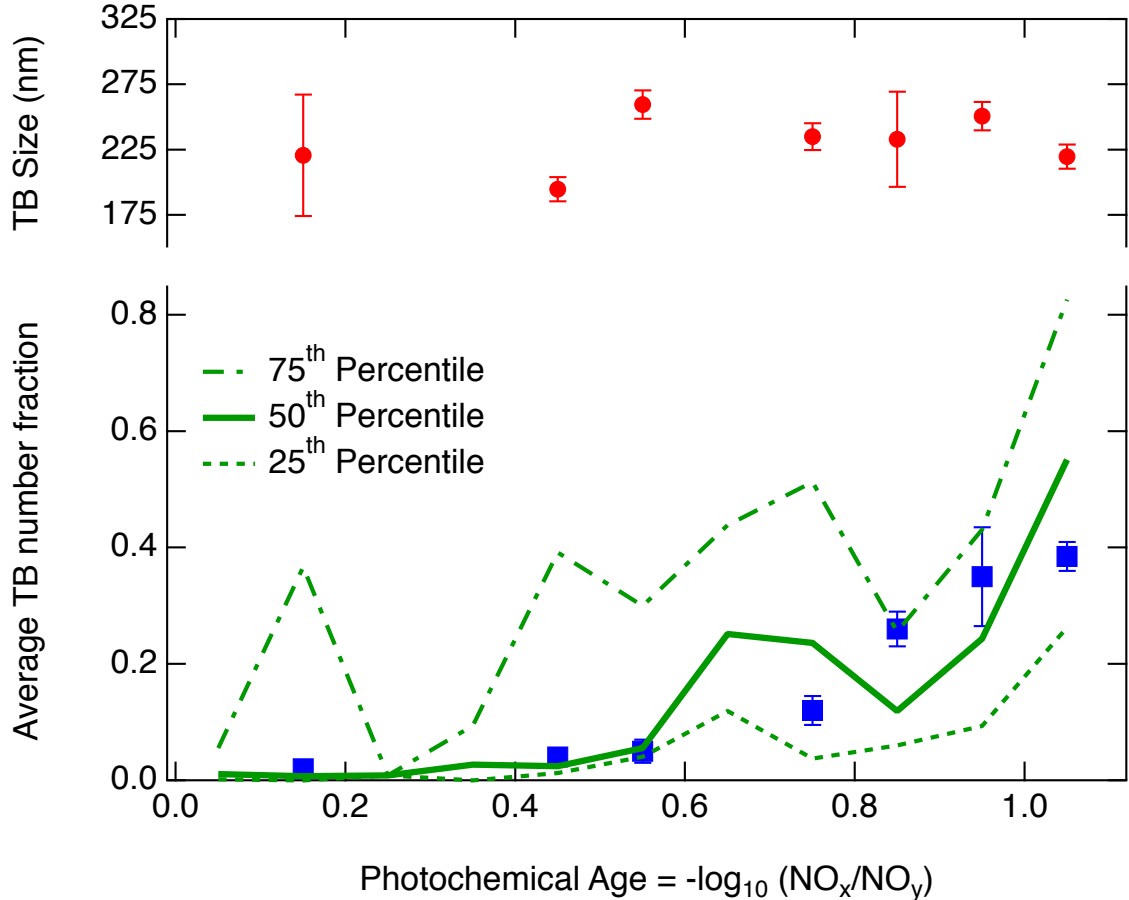

**Figure 6: The 25[th], 50[th], and 75[th] percentile lines (green dotted, solid, and dot-dashed lines, respectively) represents the TB number fraction (167 analyzed TEM sample grids) as a function of photochemical age for 11 BBOP wildfire flights conducted in the Pacific Northwest and demonstrates that TB formation occurs downwind from the source. The spread described by the 25[th] and 75[th] percentiles from individual data points (dotted and dot-dashed green lines) is believed to be due to TEM samples often included background air with varying properties along with the plume being investigated. Blue squares are the mean number fractions for the Colockum Tarps fire. Upper trace (red circles) shows that the TB average diameter is independent of photochemical age for the Colockum Tarps fire, consistent with TBs being processed primary particles. Error bars are one standard deviation.**



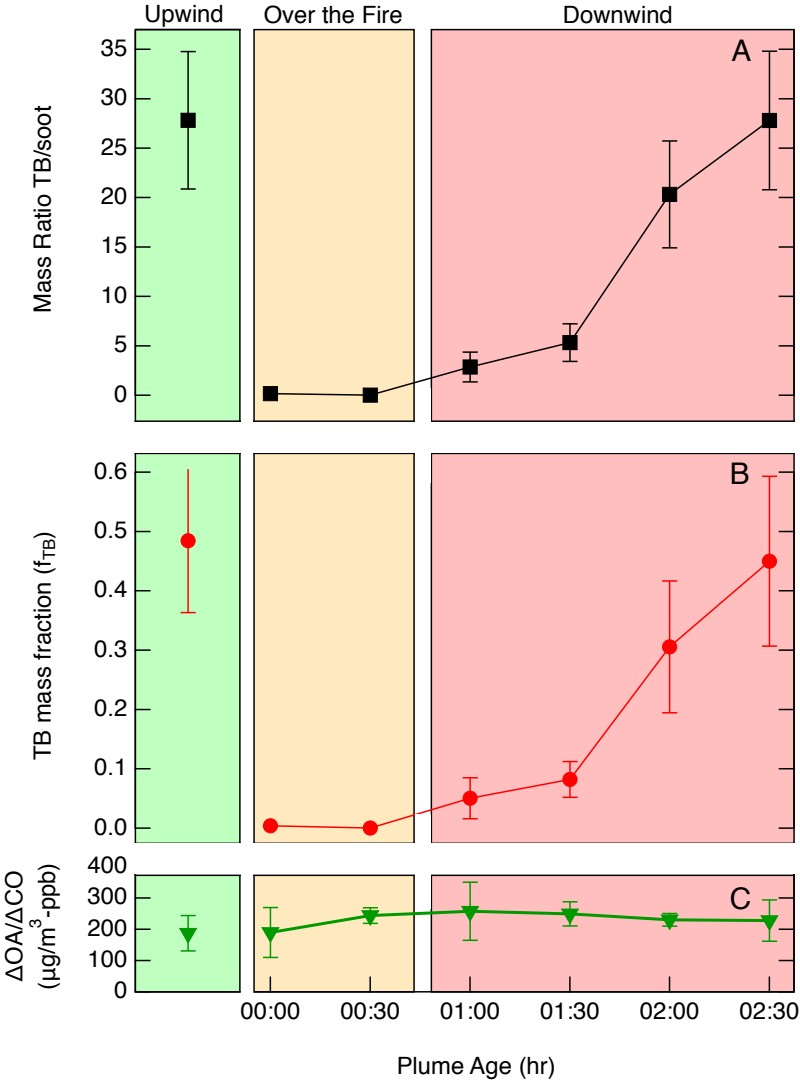

**Figure 7a: Mass ratio of TB to ns-soot (black squares) as a function of plume age (based on 953 particles). Ratios are the average TB volume-equivalent diameters (VEDs) from TEM samples collected at given plume age to the average ns-soot VEDs derived from the SP2. Background TB/ns-soot ratio was measured during upwind transect. Figure 7b: Mass fractions of TBs (red circles) in the smoke plume as a function of plume age are derived from TEM TB/ns-soot mass ratios, rBC mass loadings from SP2, and NR-PM from SP-AMS. As the wildfire plume ages, the TB mass fractions increase from 0 to 0.45 of the total aerosol mass loading. Upwind of the Colockum Tarps fire, the background TB mass fraction is estimated to be about ~50% (green shaded region). The error bars in Fig. 6b represent the measurement precisions (1σ); the estimated uncertainties in $f_{TB}$ is 61%. Figure 7c: Organic aerosol (OA) loading ratioed against CO – to correct for dilution – for background (upwind) and as a function of plume age.**



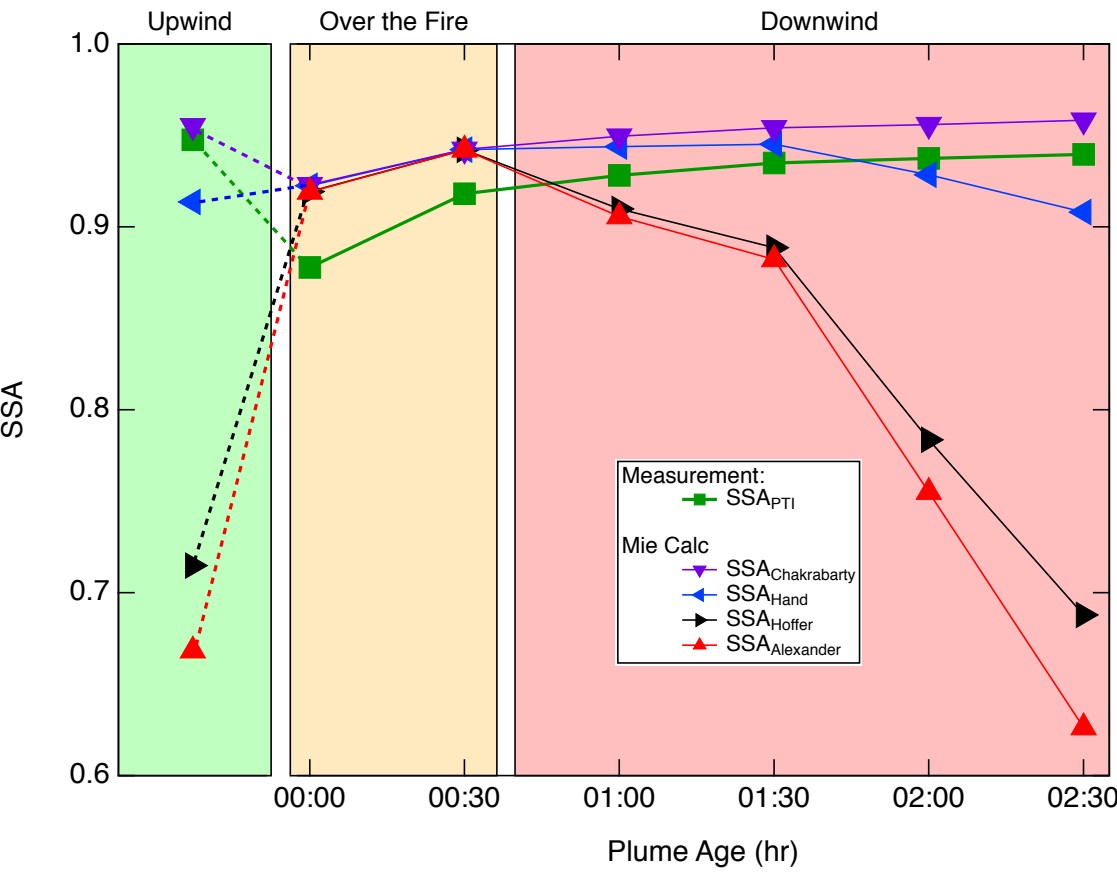

**Figure 8: Single-scattering albedo (SSA) derived assuming four refractive indices for TBs (SSA$_{Alexander}$ − red, upward pointing triangles [1.67 − 0.27i; Alexander et al., 2008); SSA$_{Hoffer}$ − black, right facing triangles [1.84 − 0.21i; Hoffer et al., 2016]; SSA$_{Hand}$ − blue, left facing triangles [1.56 − 0.02i; Hand et al., 2005] and SSA$_{Chakrabarty}$ − purple, downward pointing triangles [1.80 − 0.007i; Chakrabarty et al., 2010]). Comparison of the calculated SSAs with that derived from primary measurements of aerosol absorption and scattering reveal that TBs observed in the field by *in situ* measurements (Hand et al., 2005; Chakrabarty et al., 2010) do not exhibit the strong absorption suggested by laboratory studies (Alexander et al; 2008; Hoffer et al., 2016) (red upward pointing triangles and black right facing triangles.**