# Peer review of "Formation and evolution of Tar Balls from Northwestern US wildfires"

_Atmospheric Chemistry and Physics, 2018_

## Referee Comment (RC1) · Anonymous Referee #1 · 10 Mar 2018

This manuscript characterizes the microphysical properties of tar balls sampled during the BBoP campaign. This is an important finding and deserves to be published. I would recommend publication of this manuscript after mandatory revision. Below are my major comments:

1) In the abstract, the sentence "Brown carbon is a poorly characterized mixture that includes tar balls (TBs)" conveys a very vague meaning and needs to be edited or removed. I suggest not to mention the word "brown carbon" here. Just defining tar balls should suffice.

2) I like that this paper finally tackles the question of how tar balls are actually made. I like the terminology "processed primary particles". This needs to be mentioned in the abstract. A sentence or two should also be added to the abstract on how these

particles differ in composition and optical properties from those generated using heat-shock treatment in the laboratory by Hoffer et al.

3) The authors determine aging using the NOX/NOY metric, which is fine. However, by parameterizing aging using this metric will make it difficult to place their findings within the accepted context of the atmospheric chemistry community. I recommend a Van Krevelen (VK) diagram if they do have the necessary data from the AMS. I think adding a VK plot would enable better comparison of this dataset with other studies.

4) If the authors have optical measurements and size distribution data available, why didn't they just inverse Mie calculations to derive complex m rather than perform forward calculations of literature data? The consensus is that m varies according to fuel type and burn conditions, and using a single value for a given fuel is an incomplete basis for comparison. For example, assuming that all Alaskan duff particles have m=1.75-0.002i is neither reasonable nor rigorous.

5) Continuing the discussion of complex m, Sumlin et al. (2017 and 2018) have shown that m associated with the organic fraction of carbonaceous aerosol varies widely as a function of burn conditions and photochemical age. This work further emphasizes that it is preferable to do inverse Mie calculations rather than rely on previously reported values.

Sumlin, B. J.; Pandey, A.; Walker, M. J.; Pattison, R. S.; Williams, B. J.; Chakrabarty, R. K., Atmospheric Photooxidation Diminishes Light Absorption by Primary Brown Carbon Aerosol from Biomass Burning. Environ. Sci. Tech. Let. 2017, 4 (12), 540-545.

Sumlin, B. J.; Heinson, Y. W.; Shetty, N.; Pandey, A.; Pattison, R. S.; Baker, S.; Hao, W. M.; Chakrabarty, R. K., UV-Vis-IR spectral complex refractive indices and optical properties of brown carbon aerosol from biomass burning. J. Quant. Spectrosc. Radiat. Transfer 2018, 206, 392-398.

6) Figure 5 shows TEM images of aggregates of tar balls. This is very interesting.

Chakrabarty et al. (2016; ACP) have also observed aggregates of spheres from Alaskan Peat combustion. A paragraph is warranted on how the optical parameters (SSA etc.) would differ if aggregate morphology is accounted for in the calculations. Would schemes such as Rayleigh-Debye-Gans be appropriate?

---

## Referee Comment (RC2) · Anonymous Referee #2 · 18 Mar 2018

In this paper, Sedlacek et al. present a set of measurements and analysis to understand formation and properties of tar balls (TB) in biomass burning plumes on the US. The paper is very well written, and for the most part, clear. Considering the difficulty in extracting specific information on different particles from biomass burning plumes, the papers present a set of very compelling new data on the TB physical properties and their evolution. I, therefore, strongly support the publication of these results, as I think they will be very useful to the community. I only request a few minor clarifications and small changes as discussed next.

General comments:

1. In the abstract, the authors suggest that the index of refraction found for the TBs is m=1.56-0.02i, but I did not find this value to be really discussed in the paper. The

last figure of the paper shows a comparison with different index of refraction values published in the literature. The use of the Chakrabarty et al. and the Hand et al. index of refractions seems to bracket the measured values, but why was the Hand value for k=0.02 chosen for the abstract is not clear to me. In addition, which one of the two Chakrabarty et al. values given in table 2 were used in the last figure? My point is that the abstract seems to imply a rather specific index of refraction value, while I think the range between the two works mentioned above probably would be a more realistic assertion.

2. I would consider some small reorganization or at least some referencing to specific sections in some cases where some concepts/quantities are mentioned before an explanation is given on how they are calculated or defined. An example is the TB mass fraction, but I will mention that more directly in the specific comments next.

Specific comments:

- Line 69: is this a commercial impactor?

- Line 76: "had the same qualitative trends...", in what sense? I do not fully understand this statement.

- Line 100: Why is the SP-AMS calibrated with regal black and the SP2 with fullerene? Does it matter that two different types of particles are used for calibration?

- Line 115: "using the laser-off CE = 0.5 as the presumed ambient NR-PM loadings" maybe a verb is missing as in: "using the laser-off CE = 0.5 to calculate the presumed ambient NR-PM loading?

- Line 125: "enhancement" with respect to the background? How was the background determined?

- Line 171: This is an example of when a concept, I believe, is discussed before it is defined. The question the reader might have is: how was the mass ratio calculated? This is discussed later, but here is confusing. Either postpone this discussion, or define

the ratio before, or at least refer to the section where the mass ratio is clearly defined. The same issue appears in line 182.

- End of sentence on line 188: I think they refer to figure 6.

- Line 206: I think it would be useful to explain how R is calculated, given that what was measured (at least reported earlier) is the number fraction. To calculate the mass ratio a few assumptions (e.g., the bulk mass density) and different measurement of the soot morphology especially needs to be made, I would guess. Again, this becomes clear below, but here is a bit obscure. I would move the mass ratio definition earlier on, or at least I will clearly refer here to the section where it is defined.

- Line 248: "The results presented here raise the question as to how best to describe TBs." One "to" too many, I believe.

- Line 274: Maybe refer back to equation 1a so it is clear what the four components are.

- Lines 275-276: It would be good to have error bars associated with the SSA estimates from the measurements. In addition, I guess Mie was used for soot as well, which might underestimate the SSA of that component, although maybe that's not a real issue owing to the fact that the Mie underestimation is mostly in the scattering.

- Figure 4: y-axis label, "Age" seems an odd term here. I would think "Age" should have dimensions of time, while the quantity plotted is probably dimensionless. Same on the x-axis label of figure 6, although in this case there is the term "photochemical" in front, at least.

- Figure 8: As mentioned earlier, error bars on the measured data (green curve) would help interpret which literature value(s) of the index of refraction is(are) consistent with the measurements. Caption, for Chakrabarty et al. in table 2, are reported two different index of refraction values, which one was used here? In addition, why one or the other?

[Figure]

[Figure]

---

## Author Comment (AC1) · 12 May 2018

Anonymous Referee #1 This manuscript characterizes the microphysical properties of tar balls sampled during the BBoP campaign. This is an important finding and deserves to be published. I would recommend publication of this manuscript after mandatory revision. Below are my major comments:

1) In the abstract, the sentence "Brown carbon is a poorly characterized mixture that includes tar balls (TBs)" conveys a very vague meaning and needs to be edited or removed. I suggest not to mention the word "brown carbon" here. Just defining tar balls should suffice.

We agree that this sentence was somewhat confusing. We have changed the sentence

to read, "Tar balls (TBs) are a type of brown carbonaceous particle apparently unique to biomass burning." We have chosen to leave the term "brown carbonaceous particle" in the definition of TBs, as TBs absorb light as been well-established in the literature (e.g., our citations, Chakrabarty et al, 2010 and 2016, Alexander et al., 2008, and Hoffer et al., 2016). In this work, we tested three different TB refractive indices (RI) and found that the RI's that gives the best agreement with that derived SSA values from our measurements were consistent with TBs exhibiting weak light absorption (i.e., m similar to the literature values 1.56 – 0.02i or 1.56 – 0.02i).

2) I like that this paper finally tackles the question of how tar balls are actually made. I like the terminology "processed primary particles". This needs to be mentioned in the abstract. A sentence or two should also be added to the abstract on how these particles differ in composition and optical properties from those generated using heat-shock treatment in the laboratory by Hoffer et al.

We thank the reviewer for the encouraging words. We spent many hours discussing the pros and cons of labeling tar balls as "processed primary particles". We agree that this terminology should be included in the abstract. We have added the following sentence: "Given the observed evolution of TBs it is recommended that these particles be labeled as processed primary particles, thereby distinguishing TBs formation/evolution from secondary organic aerosols."

As for the reviewer's comments about the laboratory results, we unfortunately cannot meet the request at this time. These initial laboratory experiments were preliminary and focused on attempting to determine a reasonable measure of the collection efficiency of the SP-AMS to laboratory generated tar balls. We accomplished this narrow task. We did not collect optical measurements in the laboratory. Furthermore, given the preliminary nature of these laboratory experiments, we are still working on interpreting any SP-AMS chemical information obtained during these studies. This information, while interesting, remains outside the scope of this work.

3) The authors determine aging using the NOX/NOY metric, which is fine. However, by parameterizing aging using this metric will make it difficult to place their findings within the accepted context of the atmospheric chemistry community. I recommend a Van Krevelen (VK) diagram if they do have the necessary data from the AMS. I think adding a VK plot would enable better comparison of this dataset with other studies.

We thank the reviewer for the relevant suggestion. The requested VK diagram for the second flight on 30 July is shown here, colored by the HROrg mass loading concentrations to differentiate between sampled plumes (high loadings) and background air (low loadings). These data were all collected with both the laser and tungsten vaporizers on. The elemental ratios have been estimated using the Aiken et al. (2008) method and the Canagaratna et al. (2015) corrections for SP-AMS laser vaporizer data have been applied. As expected, the organic dominated biomass burning particles are less oxidized (lower O:C and higher H:C) when sampled nearest to the fire (highest loadings) and more oxidized (higher O:C and lower H:C) downwind and in the background air (lowest loadings).

VK diagram insert here

While we agree with the reviewer that this information is highly interesting, we have chosen not to include it here for several reasons. First, age dependent changes in elemental ratios from biomass burns are not as well characterized or understood as that from other emission sources. Second, BBOP represented the first research flights for the SP-AMS operated with dual laser and thermal vaporizers. This instrument configuration is not as well characterized as a standard AMS. Comparisons with other studies would have to take into account the possibility of differences in instrumentation. For example, the Canagaratna et al. (2015) SP-AMS laser vaporizer corrections were obtained from laboratory studies of various organic compounds on refractory black carbon particles vaporized by the laser vaporizer, whereas the BBOP samples were for a highly externally mixed aerosol with significant mass spectral ion signals coming from both the laser and tungsten vaporizers. We are currently working on understanding the

instrumental effects on these measurements, through comparing laser on and off measurements for sequential plumes transects (which were done for this explicit purpose during BBOP) and independent laboratory studies, and will present these results in the future in another form.

Aiken, A.C., DeCarlo, P.F., Kroll, J.H., Worsnop, D.R., Huffman, J.A., Docherty, K.S., Ulbrich, I.M., Mohr, C., Kimmel, J.R., Sueper, D., Sun, Y., Zhang, Q., Trimborn, A., Northway, M., Ziemann, P.J., Canagaratna, M.R., Onasch, T.B., Alfarra, M.R., Prevot, A.S.H., Dommen, J., Duplissy, J., Metzger, A., Baltensperger, U., and Jimenez, J.L. (2008). O/C and OM/OC Ratios of Primary, Secondary, and Ambient Organic Aerosols with High-Resolution Time-of-Flight Aerosol Mass Spectrometry. Environ. Sci. Technol., 42(12):4478–4485.

Canagaratna, M.R., Massoli, P., Browne, E.C., Franklin, J.P., Wilson, K.R., Onasch, T.B., Kirchstetter, T.W., Fortner, E.C., Kolb, C.E., Jayne, J.T., Kroll, J.H., and Worsnop, D.R. (2015). Chemical Compositions of Black Carbon Particle Cores and Coatings via Soot Particle Aerosol Mass Spectrometry with Photoionization and Electron Ionization. J. Phys. Chem. A, 119(19):4589–4599

4) If the authors have optical measurements and size distribution data available, why didn't they just inverse Mie calculations to derive complex m rather than perform forward calculations of literature data? The consensus is that m varies according to fuel type and burn conditions, and using a single value for a given fuel is an incomplete basis for comparison. For example, assuming that all Alaskan duff particles have m=1.75-0.002i is neither reasonable nor rigorous.

Aerosol particle concentrations in wildfire plumes greatly exceeded the coincidence design limits of the UHSAS and PCASP probes that were used to determine size distributions in the size range relevant to scattering. Flow control to the UHSAS was not adequate to reduce particle concentrations to acceptable values. The UHSAS particle-by-particle data was not consistent with random particle arrival times and thus was of

no help in coincidence-correcting the data. The inadequacy of the particle measurements was demonstrated via Mie calculations in smoke plumes in which scattering based on UHSAS or PCASP size distributions were of order 10% to 50% of that measured with a TSI nephelometer. The variability and magnitude of the discrepancy are quite outside of the range that can be generated by a variable refractive index.

The intent of Figure 8 was not to retrieve a refractive index, but rather to show that our scattering and absorption measurements are consistent with literature values for the refractive index of weakly absorbing TBs, but not with strong absorbers. Reasonable changes in the real component of the refractive index of TBs or organics do not change this conclusion.

5) Continuing the discussion of complex m, Sumlin et al. (2017 and 2018) have shown that m associated with the organic fraction of carbonaceous aerosol varies widely as a function of burn conditions and photochemical age. This work further emphasizes that it is preferable to do inverse Mie calculations rather than rely on previously reported values. Sumlin, B. J.; Pandey, A.; Walker, M. J.; Pattison, R. S.; Williams, B. J.; Chakrabarty, R. K., Atmospheric Photooxidation Diminishes Light Absorption by Primary Brown Carbon Aerosol from Biomass Burning. Environ. Sci. Tech. Let. 2017, 4 (12), 540-545. Sumlin, B. J.; Heinson, Y. W.; Shetty, N.; Pandey, A.; Pattison, R. S.; Baker, S.; Hao, W. M.; Chakrabarty, R. K., UV-Vis-IR spectral complex refractive indices and optical properties of brown carbon aerosol from biomass burning. J. Quant. Spectrosc. Radiat. Transfer 2018, 206, 392-398.

The retrievals and retrieval software described by Sumlin et al. (2017 and 2018) are impressive. High quality data is a must in such retrievals. Unfortunately, as above, we do not have the data to pursue similar calculations for BBOP. Even if all of the instruments worked as intended, an inversion of size and scattering data to yield a refractive index would be difficult. For example, Sumlin et al. (2017 and 2018) averaged their data over 5 minutes. The plumes we sampled had significant structure within one km (10-second integration time at a sampling speed of 100 m/s), thereby not allowing

long averaging times.

6) Figure 5 shows TEM images of aggregates of tar balls. This is very interesting. Chakrabarty et al. (2016; ACP) have also observed aggregates of spheres from Alaskan Peat combustion. A paragraph is warranted on how the optical parameters (SSA etc.) would differ if aggregate morphology is accounted for in the calculations. Would schemes such as Rayleigh-Debye-Gans be appropriate?

We thank the reviewer for bringing the TB aggregates in our TEM images to our attention. The authors agree that the observation of aggregated TBs is interesting and potentially provides insight into their formation (e.g., the TBs number concentration was high enough to drive TBs coagulation). We have added the following sentence at line 159 to reflect corroboration of our observations with that reported by Chakrabarty et al., (2016). "Similar to that reported by Chakrabarty et al., (2016), agglomerated TBs are observed in some TEM images." The appropriate citation has been added to the reference list (line 346 in the original manuscript).

With respect to the impact that agglomerated TBs would have on optical properties, this is outside the scope of the current work. First, it is not clear how aggregated TBs may affect the optical properties. In contrast to soot particles, where very small spheres aggregate into larger particles that increase light scattering, aggregated ~200 nm TBs may actually exhibit decreased scattered light (i.e., lower SSA) due to the decreasing scattering efficiency for larger sized particles in the visible. Second, we have not quantified the number concentrations of the aggregated TBs nor the number of TBs in each aggregate, which would have to be done prior to addressing this specific comment. Third, TBs particles are too large (~200 nm) to use the Rayleigh-Debye-Gans approximation (Sorensen, 2001, Aerosol Science and Tech), leaving a T-matrix or discrete dipole approximation (DDA) calculation as a method to explore this impact. Further work will be required to adequately address this intriguing comment by the reviewer.

Please also note the supplement to this comment:
https://www.atmos-chem-phys-discuss.net/acp-2018-41/acp-2018-41-AC1-supplement.pdf

[Figure]

**Fig. 1.**

[Figure]

---

## Author Comment (AC2) · 12 May 2018

In this paper, Sedlacek et al. present a set of measurements and analysis to understand formation and properties of tar balls (TB) in biomass burning plumes on the US. The paper is very well written, and for the most part, clear. Considering the difficulty in extracting specific information on different particles from biomass burning plumes, the papers present a set of very compelling new data on the TB physical properties and their evolution. I, therefore, strongly support the publication of these results, as I think they will be very useful to the community. I only request a few minor clarifications and small changes as discussed next.

General comments: 1. In the abstract, the authors suggest that the index of refraction

found for the TBs is m=1.56-0.02i, but I did not find this value to be really discussed in the paper. The last figure of the paper shows a comparison with different index of refraction values published in the literature. The use of the Chakrabarty et al. and the Hand et al. index of refractions seems to bracket the measured values, but why was the Hand value for k=0.02 chosen for the abstract is not clear to me. In addition, which one of the two Chakrabarty et al. values given in table 2 were used in the last figure? My point is that the abstract seems to imply a rather specific index of refraction value, while I think the range between the two works mentioned above probably would be a more realistic assertion.

The authors agree with the Reviewer. Our intention is to show that the BBOP data are consistent with TBs having a refractive index with a low imaginary component, but not consistent with the two high imaginary components in the literature. Abstract wording (line 18) changed to read: "Mie calculations are consistent with weak light absorbance (i.e., m similar to the literature values $1.56 - 0.02i$ or $1.80 - 0.007i$)." Similar changes have been made in text (e.g., we added "or $1.80 - 0.007i$" to line 276).

2. I would consider some small reorganization or at least some referencing to specific sections in some cases where some concepts/quantities are mentioned before an explanation is given on how they are calculated or defined. An example is the TB mass fraction, but I will mention that more directly in the specific comments next.

We have carefully read through the manuscript and, with the help of the both Reviewers, identified locations where action was needed (e.g., line 171).

Specific comments:
- Line 69: is this a commercial impactor?

Yes. The impactor sampler was made by Arios Inc., Tokyo, Japan. Text has been added to clarify this.

- Line 76: "had the same qualitative trends. . .", in what sense? I do not fully understand this statement.

found for the TBs is m=1.56-0.02i, but I did not find this value to be really discussed in the paper. The last figure of the paper shows a comparison with different index of refraction values published in the literature. The use of the Chakrabarty et al. and the Hand et al. index of refractions seems to bracket the measured values, but why was the Hand value for k=0.02 chosen for the abstract is not clear to me. In addition, which one of the two Chakrabarty et al. values given in table 2 were used in the last figure? My point is that the abstract seems to imply a rather specific index of refraction value, while I think the range between the two works mentioned above probably would be a more realistic assertion.

The authors agree with the Reviewer. Our intention is to show that the BBOP data are consistent with TBs having a refractive index with a low imaginary component, but not consistent with the two high imaginary components in the literature. Abstract wording (line 18) changed to read: "Mie calculations are consistent with weak light absorbance (i.e., m similar to the literature values $1.56 - 0.02i$ or $1.80 - 0.007i$)." Similar changes have been made in text (e.g., we added "or $1.80 - 0.007i$" to line 276).

2. I would consider some small reorganization or at least some referencing to specific sections in some cases where some concepts/quantities are mentioned before an explanation is given on how they are calculated or defined. An example is the TB mass fraction, but I will mention that more directly in the specific comments next.

We have carefully read through the manuscript and, with the help of the both Reviewers, identified locations where action was needed (e.g., line 171).

Specific comments:
- Line 69: is this a commercial impactor?

Yes. The impactor sampler was made by Arios Inc., Tokyo, Japan. Text has been added to clarify this.

- Line 76: "had the same qualitative trends. . .", in what sense? I do not fully understand this statement.

The authors agree that the wording is confusing. The sentence has been modified to read: "Ns-soot volumes as determined from 2D TEM images using fractal parameters from Adachi et al (2010) were in agreement with that derived from SP2 measurements of refractory black carbon (rBC)."

- Line 100: Why is the SP-AMS calibrated with regal black and the SP2 with fullerene? Does it matter that two different types of particles are used for calibration?

The reviewer asks a very good question. First, however, we will point out that the data used from the SP-AMS in this work derives from the ammonium nitrate calibration of non-refractory particulate matter (NR-PM), here dominated by organic aerosol particles, and a relative collection efficiency obtained by comparing the laser on NR-PM to the laser off NR-PM. Under SP-AMS laser off conditions, the SP-AMS operates as a standard AMS with a resistively heated tungsten vaporizer, which is calibrated with ammonium nitrate particles (Canagaratna et al., 2007). Under SP-AMS laser on conditions, the dominant NR-PM signals in the biomass burning plumes were still derived from the standard tungsten vaporizer. Direct comparisons between the two sampling modes indicate that the laser on mode increased the measured NR-PM organic mass loading due to several factors, including (1) different collection efficiencies between the two different vaporizers, (2) inadvertent heating of the ion formation chamber and tungsten vaporizer above their typical values, and (3) potential different sensitivities between ions formed from the two vaporizers (Onasch et al., 2012; Willis et al., 2014; Lee et al., 2015). Given the incomplete understanding of these different influential factors, we determined and applied separate collection efficiency (CE) factors for the NR-PM signal obtained during laser on (0.76) and laser off (0.5) modes.

Getting back to Reviewer #2 specific question, given the lack of a comprehensive understanding of black carbon particle chemical and physical properties, it is very common for different measurement techniques to rely on different calibration standards. In the current case, Onasch et al., (2014, 2015) and Corbin et al. (2015) explored the refractory carbon cluster ion distributions measured for different black carbon particle

types, including Regal black and fullerene soot, which are manufactured carbon blacks, and atmospherically relevant soots from diesel and biomass burning. Both the diesel and the biomass burning ion distributions were more closely replicated by Regal black particles than fullerene soot particles. The major differences here pertain to Cx+ ion signals for clusters greater than C5+; fullerene soot generates significant large carbon cluster ions, whereas Regal black does not exhibit these ion signals.

The amount of incandescent light collected by a SP2 instrument also varies by soot particle type. As has been discussed in the literature (Moteki and Kondo, 2010; Gysel et al., 2011; Laborde et al., 2012; Baumgardner et al., 2012), fullerene soots tend to give calibration curves that better represent ambient soot particles than particles like Regal black. Therefore, within the SP2 community, fullerene is widely used as a calibration standard (Baumgardner et al., 2012). These differences relate to the amount of incandescence light collected for a given mass of particles. More graphitic or refractory particles, such as graphite and carbon blacks heat to higher temperatures prior to vaporization/oxidation than either fullerene soot or ambient soots.

As indicated above, these differences in black carbon calibrations are interesting and complex, but do not directly relate to the results presented here.

Canagaratna, M.R., Jayne, J.T., Jimenez, J.L., Allan, J.D., Alfarra, M.R., Zhang, Q., Onasch, T.B., Drewnick, F., Coe, H., Middlebrook, A., Delia, A., Williams, L.R., Trimborn, A.M., Northway, M.J., DeCarlo, P.F., Kolb, C.E., Davidovits, P., and Worsnop, D.R. (2007). Chemical and microphysical characterization of ambient aerosols with the aerodyne aerosol mass spectrometer. Mass Spectrom. Rev., 26(2):185–222. Onasch, T.B., Trimborn, A., Fortner, E.C., Jayne, J.T., Kok, G.L., Williams, L.R., Davidovits, P., and Worsnop, D.R. (2012). Soot Particle Aerosol Mass Spectrometer: Development, Validation, and Initial Application. Aerosol Sci. Technol., 46(7):804–817. Willis, M.D., Lee, A.K.Y., Onasch, T.B., Fortner, E.C., Williams, L.R., Lambe, A.T., Worsnop, D.R., and Abbatt, J.P.D. (2014). Collection efficiency of the soot-particle aerosol mass spectrometer (SP-AMS) for internally mixed particulate black carbon. Atmos. Meas.

Tech., 7(12):4507–4516. Lee, A.K.Y., Willis, M.D., Healy, R.M., Onasch, T.B.B., and Abbatt, J.P.D. (2015). Mixing state of carbonaceous aerosol in an urban environment: single particle characterization using the soot particle aerosol mass spectrometer (SP-AMS). Atmos. Chem. Phys., 15(4):1823–1841. Corbin, J.C., Sierau, B., Gysel, M., Laborde, M., Keller, A., Kim, J., Petzold, A., Onasch, T.B., Lohmann, U., and Mensah, a. a. (2014). Mass spectrometry of refractory black carbon particles from six sources: carbon-cluster and oxygenated ions. Atmos. Chem. Phys., 14(5):2591–2603. Onasch, T.B., Trimborn, A., Fortner, E.C., Jayne, J.T., Kok, G.L., Williams, L.R., Davidovits, P., and Worsnop, D.R. (2012). Soot Particle Aerosol Mass Spectrometer: Development, Validation, and Initial Application. Aerosol Sci. Technol., 46(7):804–817. Onasch, T.B., Fortner, E.C., Trimborn, A.M., Lambe, A.T., Tiwari, A.J., Marr, L.C., Corbin, J.C., Mensah, A. a., Williams, L.R., Davidovits, P., and Worsnop, D.R. (2015). Investigations of SP-AMS Carbon Ion Distributions as a Function of Refractory Black Carbon Particle Type. Aerosol Sci. Technol., 49(6):409–422. Moteki, N. and Kondo, Y. (2010) Dependence of Laser-Induced Incandescence on Physical Properties of Black Carbon Aerosols: Measurements and Theoretical Interpretation, Aerosol Sci. Technol., 44, 663–675. Gysel, M., Laborde, M., Olfert, J. S., Subramanian, R., and Grohn, A. J. (2011) Effective density of Aquadag and fullerene soot black car- bon reference materials used for SP2 calibration, Atmos. Meas. Tech., 4, 2851–2858, doi:10.5194/amt-4-2851-2011. Laborde, M., Mertes, P., Zieger, P., Dommen, J., Baltensperger, U., and Gysel, M. (2012) Sensitivity of the Single Particle Soot Photometer to different black carbon types, Atmos. Meas. Tech., 5, 1031– 1043, doi:10.5194/amt-5-1031-2012. Baumgardner, D., Popovicheva, O., Allan, J., Bernardoni, V., Cao, J., Cavalli, F., Cozic, J., Diapouli,  E., Eleftheriadis, K., Genberg, P. J., Gonzalez, C., Gysel, M., John, A., Kirchstetter, T. W., Kuhlbusch, T. A. J., Laborde, M., Lack, D., Muller, T., Niessner, R., Petzold, A., Piazzalunga, A., Putaud, J. P., Schwarz, J., Sheridan, P., Subramanian, R., Swietlicki, E., Valli, G., Vecchi, R., and Viana, M. (2012) Soot reference materials for instrument calibration and intercomparisons: a workshop summary with recommendations Atmos. Meas. Tech., 5, 1869–1887

- Line 115: "using the laser-off CE = 0.5 as the presumed ambient NR-PM loadings" maybe a verb is missing as in: "using the laser-off CE = 0.5 to calculate the presumed ambient NR-PM loading?

We agree that the wording was awkward. We have changed the sentence to read, "Counting each transect pair as a data point and using the laser-off mode corrected with CE = 0.5 as the best estimate of ambient NR-PM loading, we obtain an average laser-on mode CE = 0.76 ($1\sigma$ = 0.10) with CE values that range from 0.63 to 0.88 for a given transect pair."

- Line 125: "enhancement" with respect to the background? How was the background determined?

The background, which was readily apparent, was determined during sampling times in between plumes. These small changes in background mattered little for most in-plume measurements as we sampled close to the fires during BBOP and the concentrations were typically much higher than in background surround air.

- Line 171: This is an example of when a concept, I believe, is discussed before it is defined. The question the reader might have is: how was the mass ratio calculated? This is discussed later, but here is confusing. Either postpone this discussion, or define the ratio before, or at least refer to the section where the mass ratio is clearly defined. The same issue appears in line 182.

We appreciate the Reviewer's comment about using a term/concept before formally defining it and how this could create difficulties. We have changed in line 176 the text "TB/ns-soot mass " to "TB/ns-soot number ratio". In responding to this comment, it became clear that we should have used the number ratio instead of mass ratio given that we were discussing the TB/ns-soot number ratio in this paragraph.

In line 182, the sentence currently reads: "The downwind increase in TB mass is due to an increase in number concentration rather than a change in particle size (green 25th,

50th, and 75th percentile lines in the lower panel of Figure 6)." This sentence is merely saying that an increase in TB mass contribution in the plume is due not to individual particle size growth, but instead, due to increased number of TB particles. The authors feel that the sentence is clear on this point and that there is no new concept that is being presented. Unrelated this to comment, the authors have elected to make a small stylistic change to Figure 6: use of shading to define the 25th to 75th percentile range. Manuscript text that referenced this figure has been updated to reflect this stylistic change.

- End of sentence on line 188: I think they refer to figure 6.

Yes, the Reviewer is correct, and this has been changed.

- Line 206: I think it would be useful to explain how R is calculated, given that what was measured (at least reported earlier) is the number fraction. To calculate the mass ratio a few assumptions (e.g., the bulk mass density) and different measurement of the soot morphology especially needs to be made, I would guess. Again, this becomes clear below, but here is a bit obscure. I would move the mass ratio definition earlier on, or at least I will clearly refer here to the section where it is defined.

We defined R as the ratio of TB mass to ns-soot mass. We have added (highlighted in bold) the following to the end of the sentence at line 206: "TB fractional contribution to total mass is denoted by fTB and we define R as the ratio MTB/Msoot derived from TEM and SP2, respectively."

- Line 248: "The results presented here raise the question as to how best to describe TBs." One "to" too many, I believe.

The authors appreciate the extra care about the grammar shown by the Reviewer: the extra "to" has been removed.

- Line 274: Maybe refer back to equation 1a so it is clear what the four components are.

We have added (highlighted in bold) the following text to the end of the sentence in line 274 "…..where i labels the four aerosol components in our calculation (see equation 1a)".

- Lines 275-276: It would be good to have error bars associated with the SSA estimates from the measurements. In addition, I guess Mie was used for soot as well, which might underestimate the SSA of that component, although maybe that's not a real issue owing to the fact that the Mie underestimation is mostly in the scattering

Error bars have been added to the SSA estimates derived from the measurements.

- Figure 4: y-axis label, "Age" seems an odd term here. I would think "Age" should have dimensions of time, while the quantity plotted is probably dimensionless. Same on the x-axis label of figure 6, although in this case there is the term "photochemical" in front, at least.

The Reviewer is correct. The x-axis label has been changed to photochemical age. Defined in text as – Log10 ([NOx]/[NOy]), which is dimensionless.

- Figure 8: As mentioned earlier, error bars on the measured data (green curve) would help interpret which literature value(s) of the index of refraction is(are) consistent with the measurements. Caption, for Chakrabarty et al. in table 2, are reported two different index of refraction values, which one was used here? In addition, why one or the other?

We have added error bars to the experimentally-derived SSA estimates as discussed in response to the Reviewer's earlier comment. As highlighted in Table 3, we used at TB refractive index (RI) of 1.8 – 0.007i from Chakrabarty in our calculated SSAs. While we could have added an additional SSA curve on figure 8 – to reflect the other TB RIs – the objective of this figure was to bound our experimentally-derived SSA estimates using four published RIs whose imaginary component span two orders of magnitude.

---

## Author Response (AR2)

Inserted at line 156 "and Girotto et al., (2018)"

Inserted at line 367-369 the following reference:
Girotto, G., China, S., Bhandari, J.,  Gorkowski, K., Scarnato, B. V., Capek, T., Marinoni, A., Veghte, P.,  Kulkarn, G., Aiken, A. C., Dubey, M., and Mazzoleni C. (2018) Fractal-like Tar Ball Aggregates from Wildfire Smoke Environ. Sci. Technol. Lett. 2018, 5, 360−365